# NADPH Oxidases and Oxidative Stress in the Pathogenesis of Atrial Fibrillation

**DOI:** 10.3390/antiox12101833

**Published:** 2023-10-06

**Authors:** Roberto Ramos-Mondragón, Andrey Lozhkin, Aleksandr E. Vendrov, Marschall S. Runge, Lori L. Isom, Nageswara R. Madamanchi

**Affiliations:** 1Department of Pharmacology, University of Michigan, 1150 West Medical Center Drive, 2301 Medical Science Research Building III, Ann Arbor, MI 48109, USA; robramos@med.umich.edu (R.R.-M.); lisom@umich.edu (L.L.I.); 2Department of Internal Medicine, Frankel Cardiovascular Center, University of Michigan, Ann Arbor, MI 48019, USA; alozhkin@ipstherapeutique.com (A.L.); vendrov@med.umich.edu (A.E.V.); mrunge@med.umich.edu (M.S.R.); 3Department of Neurology, University of Michigan, Ann Arbor, MI 48109, USA; 4Department of Molecular and Integrative Physiology, University of Michigan, Ann Arbor, MI 48109, USA

**Keywords:** NADPH oxidases, NOX1, NOX2, NOX4, Rac1, ROS, paroxysmal atrial fibrillation, permanent atrial fibrillation, cardiomyocyte, mitochondrial oxidative stress, aging, arrhythmia, electrical remodeling, structural remodeling, tachypacing, reentry, cell signaling

## Abstract

Atrial fibrillation (AF) is the most common type of cardiac arrhythmia and its prevalence increases with age. The irregular and rapid contraction of the atria can lead to ineffective blood pumping, local blood stasis, blood clots, ischemic stroke, and heart failure. NADPH oxidases (NOX) and mitochondria are the main sources of reactive oxygen species in the heart, and dysregulated activation of NOX and mitochondrial dysfunction are associated with AF pathogenesis. NOX- and mitochondria-derived oxidative stress contribute to the onset of paroxysmal AF by inducing electrophysiological changes in atrial myocytes and structural remodeling in the atria. Because high atrial activity causes cardiac myocytes to expend extremely high energy to maintain excitation-contraction coupling during persistent AF, mitochondria, the primary energy source, undergo metabolic stress, affecting their morphology, Ca^2+^ handling, and ATP generation. In this review, we discuss the role of oxidative stress in activating AF-triggered activities, regulating intracellular Ca^2+^ handling, and functional and anatomical reentry mechanisms, all of which are associated with AF initiation, perpetuation, and progression. Changes in the extracellular matrix, inflammation, ion channel expression and function, myofibril structure, and mitochondrial function occur during the early transitional stages of AF, opening a window of opportunity to target NOX and mitochondria-derived oxidative stress using isoform-specific NOX inhibitors and mitochondrial ROS scavengers, as well as drugs that improve mitochondrial dynamics and metabolism to treat persistent AF and its transition to permanent AF.

## 1. AF Pathophysiology

### 1.1. Introduction

Atrial fibrillation (AF) is the most common cardiac arrhythmia in clinical practice and contributes significantly to cardiovascular morbidity and mortality [1,2]. In addition to affecting patients’ quality of life, AF increases healthcare costs by causing multiple adverse clinical outcomes, including thromboembolism and stroke. The percentage of the population with AF is rising. In the US, the prevalence of AF is expected to reach 12.1 million by 2030 [3]. AF risk increases with age, and patients over 55 are twice as likely to develop AF with each decade [4]. Age-related cardiovascular diseases, such as hypertension, myocardial infarction, and atherosclerosis, also increase the risk of AF. Men are more likely to develop AF than women [5]. Genetics also play a role, with individuals with a family history of AF having a 40% higher risk of developing the condition [6].

### 1.2. AF Classification 

According to the American Heart Association guidelines on AF management, AF can be classified as paroxysmal, recurrent, or permanent, depending on its duration [7]. If a patient experiences AF that resolves within 7 days without any intervention after the initial episode, it is classified as paroxysmal. If AF lasts for more than 30 days and requires both electrical and pharmacological interventions for cardioversion, it is classified as persistent. AF that cannot be stopped by cardioversion and lasts more than a year is considered permanent. 

### 1.3. Arrhythmogenic Mechanisms in Atrial Fibrillation

Focal activity and reentry are two mechanisms that play crucial roles in the development of AF. The primary source of focal activity in the atrium is the pulmonary veins, and eliminating this activity through pulmonary vein isolation is considered the most effective therapy for non-permanent forms of AF [8]. Triggered activity is another important mechanism that occurs when depolarization of the membrane potential generates an impulse after the upstroke phase of an action potential (AP). Early afterdepolarizations (EADs) occur when depolarizations take place during the repolarization of the action potential, while delayed afterdepolarizations (DADs) occur after complete membrane repolarization. Prolonged action potential duration (APD) makes EADs more likely to occur.

AP prolongation is mainly caused by decreased repolarizing potassium currents or increased inward sodium currents, especially late sodium (*I*_NaL_), and increased L-type Ca^2+^ currents (*I*_Ca,L_). EADs occur when voltage-gated sodium and L-type Ca^2+^ channels recover from their inactive state and reactivate during membrane repolarization. DADs, on the other hand, are associated with altered intracellular Ca^2+^ homeostasis. Spontaneous Ca^2+^ discharges from the sarcoplasmic reticulum (SR), caused by Ca^2+^ overload and dysfunction (leakiness) of ryanodine receptor 2 (RyR2), activating forward-mode action of the sodium-calcium exchanger (NCX). In this operating mode, one calcium ion is extruded from the cell in exchange for three sodium ions entering the cell. This inward sodium current (*I*_Ti_) perturbs the membrane potential, causing DADs. If the DAD reaches the threshold for sodium current activation, it can trigger a non-physiological AP. EADs and DADs are both triggered activities that are important for initiating AF in the pulmonary veins. It is notable that triggered activity has also been observed in atrial myocytes isolated from lone AF patients (AF in patients younger than 60 years without coexisting heart disease) and in AF animal models. These findings suggest that triggered activity may play a significant role in the development of AF. Further investigation is necessary to determine whether triggered atrial myocyte activity is a cause of AF [9]. 

Reentry is associated with abnormal conduction, as opposed to abnormal impulse formation. Reentry occurs when a propagating impulse fails to extinguish after the normal activation of the heart tissue and continues to re-excite the heart after the refractory period is completed. The reduced wavelength of the cardiac impulse is a determinant of reentry, which is a product of conduction time and refractory period [10,11]. Electrical remodeling, which refers to changes in the expression and function of ion channels, and structural remodeling, which refers to changes in intracellular or extracellular components, can slow conduction velocity and shorten the refractory period, thus reducing the wavelength of the cardiac impulse. Increased outward potassium current (*I*_to_) and/or decreased inward *I*_Ca,L_ can accelerate membrane repolarization and reduce the refractory period. Sodium current downregulation [12], gap junction uncoupling [13,14], atrial enlargement [15,16], and fibrosis [17,18] are other factors that can affect impulse conduction. 

Even though triggers and substrates are distinct, they are dynamic and can establish causal feedback mechanisms that favor AF progression. The mishandling of intracellular Ca^2+^ due to RyR2 dysfunction or Ca^2+^ overload, for instance, can result in the development of AF structural substrates like atrial enlargement [19]. Increased Ca^2+^ entry during atrial tachycardia triggers calcium-dependent signaling that leads to atrial hypertrophy [20]. Altered connexin expression causes fibrosis and myocyte uncoupling, slows conduction velocity, and favors reentry [18,21,22]. These intertwined arrhythmogenic mechanisms are the basis for the concept that “AF begets AF”.

In this review, we will explore how ROS (reactive oxygen species) are an important intermediary in developing AF substrates and triggers by directly oxidizing membrane ion channels, intracellular Ca^2+^ handling transporters, and structural cellular elements and/or affecting the signaling pathways that modulate them. For this purpose, we conducted a comprehensive literature review on oxidative stress, particularly NADPH oxidases (NOXs) and AF pathogenesis, using the methods below. 

We systematically searched electronic databases, such as PubMed, Google Scholar, Scopus, and Web of Science, to identify relevant studies. Our primary search used a combination of keywords and Medical Subject Headings (MeSH) terms that included reactive oxygen species, oxidative stress, NADPH oxidases, mitochondrial oxidative stress, atrial fibrillation, aging, electrical remodeling, structural remodeling, tachypacing, reentry, cell signaling, and relevant synonyms. Our inclusion criteria consisted of human and animal studies published in English, original research articles, and systematic reviews that investigated the role of ROS in atrial fibrillation pathogenesis, published on or after January 1985. We summarized and analyzed the findings from the selected studies using a narrative synthesis approach.

## 2. Sources of Cardiac ROS

Cardiovascular cells produce ROS through various enzymes such as NOX, xanthine oxidase, cyclooxygenases, lipoxygenases, myeloperoxidase, uncoupled endothelial NO synthase, and mitochondrial respiratory chain complexes [23,24]. The generation of ROS in cardiac cells is low under basal conditions [25] but can increase in response to various stimuli [25,26,27,28,29]. 

The NOX enzyme family is the most significant and the only one whose sole function is ROS production in normal and pathological conditions. The enzyme family comprises seven isoforms, including NOX 1–5 and dual oxidases (DUOX) 1 and 2 [30]. These enzymes are oxidoreductases that contain six transmembrane domains and two heme groups. They transport electrons across the membrane and reduce oxygen to superoxide (O_2_^•−^) [31,32]. The enzymes have conserved structural features, including an NADPH-binding site, a FAD-binding region, six transmembrane domains, and four highly conserved heme-binding histidines. DUOX proteins contain histidine residues in the fourth and sixth transmembrane domains. NOX enzymes are implicated in regulating various functions in the heart. NOX1, 2, 4, and 5 are found in cardiac myocytes, fibroblasts, and macrophages, and coronary vessel wall cells, including endothelial, vascular smooth muscle (VSMC), and adventitial fibroblasts.

NOX1 constitutively generates low levels of O_2_^•−^; however, increased NOX1 expression and activation produce higher levels, indicating the role of cytosolic subunits in increasing ROS production [31,33]. Optimal NOX1 activity requires plasma membrane-bound p22phox, the cytosolic subunits p47phox or NOX organizer 1 (NOXO1), p67phox or NOX activator 1 (NOXA1), and the small GTPase Rac1 [33,34,35,36]. Our research has shown that NOXA1 is the functional equivalent of p67phox in VSMCs. Interestingly, immunoreactive NOXA1 is present in intimal and medial SMCs in lesions of human early carotid atherosclerosis, which is a risk factor for AF [37]. Ang II activates renal NOXA1/NOX1-dependent ROS, enhancing tubular ENaC expression and Na^+^ reabsorption, leading to hypertension, another AF risk factor [38,39]. NOX1 and NOX4 levels were significantly increased in the myocardium, as well as in cardiomyocytes of neonates treated with doxorubicin [40], a drug that induces cardiovascular toxicity and can cause AF in mice [41] and humans [42].

New-onset AF and early AF recurrence after ablation are associated with left ventricular diastolic dysfunction (DD) [43,44,45,46,47,48,49]. Studies have demonstrated that metabolic syndrome is an independent risk factor for new-onset AF [50]. Inflammation of the microvasculature is a significant factor in DD and metabolic heart disease. Patients with DD were found to have higher levels of NOX1 [51]. This protein is expressed in vascular and immune cells and has been implicated in vascular pathology, endothelial [52] and immune cell functions [53], and myocardial remodeling in metabolic disease [51]. Inhibition of NOX1 has been suggested as a possible treatment for aging-related cardiac remodeling and HFpEF [54]. The CXCL1/CXCR2 signaling pathway is critical to immune cell trafficking and the pathogenesis of atherosclerosis, myocarditis, and other cardiovascular diseases. NOX1-derived ROS can stimulate the expression of pro-inflammatory genes, including CXCL1. The resulting action of CXCL1 in recruiting immune cells can further worsen the inflammatory response. Spontaneously hypertensive rats (SHRs) express more CXCL1 and CXCR2 mRNA in response to atrial rapid (burst) pacing [55] (Figure 1). A CXCR2 inhibitor, SB225002, significantly reduced blood pressure elevation, induction and duration of AF, atrial remodeling, macrophage recruitment, and O_2_^•−^ generation in SHRs. These effects were associated with inhibiting multiple signaling pathways, including TGF-β1/Smad2/3, NF-κB-P65, NOX1, NOX2, Kir2.1, Kv1.5, and Cx43. This suggests that NOXs may participate in macrophage recruitment and AF induction. Studies have shown a decrease in nitric oxide availability in the left atrium of swine with pacing-induced AF, indicating an oxidative stress-mediated degradation mechanism [56]. Increased production of O_2_^•−^ was observed in the left atrium and left atrial appendage, with both xanthine oxidase and NOX contributing to the increase [57]. A change in Rac1 was responsible for the increase in NOX activity, instead of a change in NOX isoform expression. 

In the heart, NOX2 and NOX4 are the most commonly found NOX isoforms. NOX2 is quiescent and found in association with p22phox in the plasma membrane. It can be activated by various stimuli such as growth factors, cytokines, G-protein coupled receptor agonists (e.g., angiotensin II and endothelin), and mechanical forces. This activation occurs through the binding and activation of cytosolic regulatory subunits (p67phox, p47phox, p40phox, and Rac1) at the plasma membrane [31,58]. NOX2 is found on sarcolemmal and t-tubule membranes of myocytes, endothelial cells, fibroblasts, macrophages, and neutrophils in the heart [58,59,60,61,62]. According to Casadei and colleagues [63], membrane-bound NOX2 is the primary source of ROS production in human atrial myocardium and myocytes. They observed a significant increase in NOX-derived ROS production in the right atrial appendages of AF patients despite no changes in the expression of p22phox or NOX2. Atrial NOX-stimulated ROS was also independently linked to an increased risk of postoperative AF [64]. 

The activation of NOX2 may cause AF, as observed in patients with paroxysmal or permanent AF [65]. In the HL-1 cell line, tachypacing induced TGF-β1 expression upregulated NOX2/4 expression and caused degradation of myofibrils [66] (Figure 1). The atria of AF patients (*n* = 4–6) had increased TGF-β expression, oxidative stress, and loss of myofibrils compared to those with sinus rhythm. The activation of NOX2 by endotoxins is implicated in the development of AF in patients with community-acquired pneumonia [67]. Obstructive sleep apnea is associated with several cardiovascular diseases, including AF [68,69]. In mice, intermittent hypoxia mimicking obstructive sleep apnea reduces and remodels atrial Cx40 and Cx43 through NOX2-dependent oxidative stress [70]. However, this study did not investigate AF pathogenesis in mice subjected to intermittent hypoxia. Rac1, NOX2, and p22phox protein levels increased significantly in goats after 2 weeks of AF and in patients with postoperative AF, without differences in leukocyte infiltration [71]. In cases of longstanding AF or atrioventricular block, the increased levels of ROS in both atria were caused by mitochondrial oxidases and uncoupled nitric oxide synthase. A study on transesophageal atrial burst pacing of NOX2 transgenic mice showed that NOX2-derived O_2_^•−^ did not affect atrial electrical and structural properties, but instead, it moderately increased the probability of AF induction [72]. These studies suggest that oxidative injury, particularly NOX2-generated oxidative injury, may play a role in AF, but do not prove that it is responsible for the genesis and maintenance of the condition [73]. A study on canine atrial myocytes and atria found that rapid pacing caused NOX2 and mitochondrial ROS-dependent oxidative injury, leading to electrical remodeling in AF. This was due to the upregulation of a constitutively active form of acetylcholine-dependent K^+^ current (*I_KAch_*) mediated by activation of PKCε, which increased in magnitude with increasing frequency of atrial tachypacing [73]. This increase in *I_KAch_* activity could increase atrial vulnerability to tachyarrhythmia in chronic AF patients, thus sustaining the condition [74,75]. 

Activation of the renin-angiotensin system has been linked to heart failure and ischemia-reperfusion-related arrhythmias. Angiotensin-converting enzyme (ACE) inhibitors and Ang II receptor blockers have been found to reduce AF [76,77]. Reduction in NOX2 activity may be a part of the mechanism [24] (Figure 1). Overexpression of ACE in mouse hearts leads to higher levels of cardiac Ang II expression and the development of spontaneous AF [78]. In rabbit cardiomyocytes, Ang II expression induces EADs by activating the NOX-ROS-calmodulin kinase II (CaMKII) pathway and enhancing *I*_Ca,L_ [79]. Oxidized CaMKII is a marker of human sinus node dysfunction (SND), and CaMKII promotes SND by enhancing sinoatrial nodal cell apoptosis. Conversely, CaMKII inhibition protects mice infused with Ang II against sinoatrial nodal cell apoptosis and SND [80]. An atrial burst pacing mouse model with transverse aortic constriction (TAC) showed increased AF susceptibility and SR Ca^2+^ leak in left atrial cardiomyocytes [81]. SR Ca^2+^ leak in pressure-overloaded hearts by RyR2 oxidation in left atrial myocytes was mediated by activation and upregulation of NOX2 and NOX4, suggesting a mechanistic relationship between SR Ca^2+^ leak induced by stretch and AF. Similarly, activation of CaMKII-dependent diastolic SR Ca^2+^ leak is observed in the right atrial myocardium of AF patients [82] and induces AF in mice [83,84]. Activation of Pak1 (p21-activated kinase 1), a downstream target in the Ang II signaling cascade, decreases Rac1-mediated NOX2 ROS production in atrial myocytes and NCX-dependent Ca^2+^ overload, inhibiting AF development [85]. However, the effects of Pak1 signaling on Ang II-induced fibrotic remodeling and AF substrate formation have not been studied. 

Nox4 forms a heterodimer with p22phox, which ensures structural stability and catalytic activity, without requiring cytosolic subunits [86,87,88]. Unlike other NOXs, NOX4 produces hydrogen peroxide (H_2_O_2_), rather than O_2_^•−^, because of an E-loop within its structure that promotes the rapid dismutation of O_2_^•−^ before it leaves the enzyme [89]. NOX4 is constitutively active and primarily regulated at its levels of expression [90]. The activity of NOX4 is also acutely regulated by oxygen tension as well as posttranslationally [91]. NOX4 is present in the endoplasmic reticulum, mitochondria, plasma membrane, and nucleus [92,93,94]. In the vasculature, endothelial cells, VSMCs, and adventitial fibroblasts express NOX4. Increased expression and activity of NOX4 have been associated with mitochondrial oxidative stress, cardiovascular dysfunction, and atherosclerosis in aged mice [94]. Moreover, increased Nox4 expression in aortic VSMCs of aged human subjects is associated with advanced atherosclerosis. Age-associated increases in NOX4 expression and activity induces a proinflammatory phenotype in VSMCs that promotes atherosclerosis [95]. Increased NOX4 expression in atherosclerotic plaque SMCs causes plaque instability through increased ROS generation, cell cycle arrest, senescence, and susceptibility to apoptosis [96]. Increased mitochondrial NOX4 levels induce vascular aging [97]. Young transgenic mice overexpressing mitochondria-targeted NOX4 displayed increased aortic stiffness, which was blunted by mitochondria-targeted catalase overexpression. By increasing the intrinsic stiffness of VSMC, remodeling the extracellular matrix, calcifying the aorta, and VSMC senescence, inflammation, and apoptosis, mitochondrial oxidative stress induces vascular aging in Nox4 transgenic mice. 

NOX4 is found in cardiomyocytes, fibroblasts, and microvascular endothelial cells in the heart [98,99,100,101,102,103,104]. The role of NOX4 in pressure overload adaptation is controversial. Increased NOX4 expression in cardiomyocytes during pressure overload can lead to mitochondrial oxidative stress, resulting in mitochondrial and cardiac dysfunction [98]. However, a study by Shah and colleagues found that overexpression of *Nox4* in cardiomyocytes can protect against chronic pressure overload by preserving myocardial capillary density [105]. The hearts of people with ischemic cardiomyopathy exhibit adverse cardiac remodeling, as well as increased levels of NOX4 and soluble epoxide hydrolase proteins [106]. Additionally, they have higher levels of CCL4 (C-C motif chemokine ligand 4, also known as macrophage inflammatory protein-1β, MIP-1β), and CCL5 (C-C motif chemokine ligand 5, also known as RANTES, regulated on activation, normal T cell expressed and secreted) compared to hearts without cardiac failure. Furthermore, there is evidence that failing heart tissue shows increased expression of full-length NOX4 [107]. 

H_2_O_2_ levels in the left atrial appendages of AF patients were more than double those in patients without AF. Additionally, H_2_O_2_ levels were found to be correlated with NOX4 levels, especially in patients with hypertension, which is a known risk factor for AF [108]. In HL-1 atrial cells treated with Ang II, a known modulator of atrial remodeling, increased levels of NOX4 and H_2_O_2_ were produced. This result implies that NOX4-derived H_2_O_2_ may play a crucial role in the development of AF (Figure 1). To further investigate the role of NOX4 in AF, Cai et al. studied the effects of *Nox4* overexpression on the cardiac phenotype of zebrafish [109]. Embryos overexpressing *Nox4* exhibit cardiac arrhythmia, increased production of O_2_^•−^ and H_2_O_2_, and redox-sensitive CaMKII activation. Attenuation of embryonic cardiac arrhythmia and oxidative stress was observed with PEG SOD and NOX4 inhibitors, while CaMKII inhibitors abolished the phenotype. These findings provide further evidence supporting the role of NOX4 in cardiac arrhythmia. 

Studies have shown that structural remodeling in AF involves fibrosis, which alters the composition and function of atrial tissue [110]. As humans age, the prevalence of atrial interstitial fibrosis increases [111,112], which has also been observed in animal models of aging [113,114], mitral regurgitation [115], and congestive heart failure [116]. Transgenic mice with cardiomyocyte overexpression of a constitutively active form of TGF-β1, MHC-TGFcys33ser, were used by Verheule et al. [18] to investigate the effects of atrial fibrosis on AF susceptibility (Figure 1). These transgenic mice showed elevated levels of TGF-β1 in both atria and ventricles, but interstitial fibrosis was only observed in atria [117]. Burst pacing led to a higher incidence of AF in transgenic mice than wild-type littermates despite no change in the atrial effective refractory period. AF vulnerability is associated with increased conduction heterogeneity in the left and decreased conduction velocity in the right atrium. Additionally, the atria have a significantly higher collagen-to-TGF-1 ratio than the ventricles [100]. The study found that NOX4 mediates atrial structural remodeling and AF vulnerability due to increased TGF-β1 responsiveness, NOX4 expression, and NOX activity in atrial fibroblasts. NOX2 expression remains unchanged.

Patients with paroxysmal and persistent/permanent AF have a larger left atrial diameter, left ventricular end-diastolic diameter, and P-wave dispersion than controls [118]. According to multivariate analysis, baseline serum NOX4 levels have a significant independent association with both types of AF, indicating that NOX4 plays a critical role in human AF pathophysiology (Figure 1). Studies have shown that intracellular Ca^2+^ overload and handling abnormalities play a significant role in AF pathogenesis [119,120,121,122]. A rise in diastolic Ca^2+^ is a critical contributor to the onset of AF by causing an increase in SR Ca^2+^ spark frequency. This leads to the initiation and spread of Ca^2+^ waves in atrial myocytes, resulting in DADs [82,119,120,121,122]. In myocytes, CaMKII is an abundant serine-threonine kinase that acts as an important ROS sensor. It may be an arrhythmogenic factor by activating downstream targets, such as RyR2 phosphorylation (Ser2814), and promoting SR leak, especially during diastole [123]. 

Rat atrial fibroblasts treated with TGF-β expressed more hyaluronan, CD44, STAT3, and collagen than ventricular fibroblasts [124] (Figure 1). Additionally, TGF-β transgenic mice and AF patients had higher levels of HA, CD44, STAT3, and collagen in their atria compared to wild-type mice and sinus rhythm subjects, respectively. Treatment of TGF-β transgenic mice with an anti-CD44 blocking antibody led to lower STAT3 and collagen expression in their atria than treatment with a control IgG antibody. Furthermore, TGF-β transgenic mice treated with anti-CD44 blocking antibodies had fewer transesophageal pacing-induced AF episodes than controls. These data suggest that the TGF-β-mediated HA/CD44/STAT3 pathway is crucial in atrial fibrosis and AF development. In HL-1 cardiomyocytes, tachypacing activation resulted in CD44-related signaling and increased expression of HA and HA synthase (HAS) [125]. However, when a HAS inhibitor or *Hsa2* siRNA was used to block HAS/HA/CD44 signaling in HL-1 myocytes, tachypacing-induced oxidative stress was reduced. This was achieved by decreasing NOX2/NOX4 expression and lowering oxidized CaMKII and phospho-RyR2 expression. In tachy-paced HL-1 myocytes and atrial tissue from people with AF, CD44 was directly associated with NOX4. *Cd44^−/−^* mice had a lower frequency of Ca^2+^ sparks, and wild-type atrial myocytes treated with anti-CD44 blocking antibody showed a diminished spark frequency. *Cd44^−/−^* mice also exhibited lower levels of oxidative stress and oxidized CaMKII-p-RyR2 in response to ex vivo tachypacing. Furthermore, they are less prone to AF induction as assessed by in vivo rapid pacing. These findings suggest that CD44/NOX4 signaling could potentially cause irregularities in Ca^2+^ handling and lead to AF pathogenesis in certain pathophysiological circumstances.

In addition to mitochondrial dysfunction [94,95,97,126,127,128,129], aging is associated with metabolic syndrome, which includes obesity [130], insulin resistance [131], and hypertension [132], all recognized risk factors for AF. In the heart, mitochondria produce approximately 95% of the heart’s cellular ATP and account for up to 30% of its volume [133]. Animal models of AF show abnormal mitochondrial structure and function [134,135,136]. Cardiomyocytes from AF patients exhibit increased mitochondrial DNA damage [137,138,139], mitochondrial structural abnormalities [140,141], and impaired function [136,137,142,143]. Mitochondrial activity disruptions can cause arrhythmogenesis by decreasing ATP availability and producing abnormal ROS levels (Figure 1) [144,145,146,147]. Brown et al. [148] discovered that mitochondrial benzodiazepine receptor ligands can prevent cardiac arrhythmias induced by glutathione oxidation by stabilizing mitochondrial membrane potential and preventing mitochondrial depolarization. Patients experiencing arrhythmia onset are affected by mitochondrial dysfunction, including reduced ability to oxidize carbohydrates and lipids, and increased sensitivity to mPTP opening. 

The formation of reentrant circuits in sustained arrhythmias is hypothesized to occur via maladaptive changes in the electrophysiological properties of myocardial tissue [129]. A shortened effective refractory period (ERP) caused by excess Ca^2+^ uptake through LTCC channels in fibrillating atria would produce such an effect. The development of reentry and shortened APDs have been linked to mitochondrial impairment and cardiac oxidative stress [149,150,151]. Na^+^ channels and connexins are major determinants of conduction [152]. Dogs subjected to chronic atrial tachycardia experienced reduced *I*_Na_, slowed conduction speed, and decreased reentry wavelengths [153]. Reductions in inward *I*_Na_ have been observed secondary to excess mitochondrial ROS production [154]. Gap junction activity and mitochondrial function are correlated [155,156], which can contribute to the slowing of conduction in certain situations. 

## 3. Oxidative Stress in the Natural History of AF

### 3.1. Role of NADPH Oxidases and Oxidative Stress in the Onset of Paroxysmal AF

After cardiac surgery, paroxysmal AF (PAF), also known as postoperative AF or POAF, commonly occurs and typically peaks between days 2 and 4 following the operation [157]. Clinical studies have shown that oxidative stress plays an essential role in the development of PAF [158,159]. Kim et al. [62] examined the sources of O_2_^•−^ production in right atrial appendage (RAA) homogenates or isolated myocytes from 15 patients with AF (six with persistent AF, four with persistent AF developing from PAF). Dipenyleneiodonium (DPI), an inhibitor of flavin-containing oxidases or apocynin, a NOX inhibitor that blocks p47phox translocation, inhibited O_2_^•−^ production in AF, suggesting a pathogenic role for NOX-derived O_2_^•−^ in AF [62]. Similarly, a borderline, yet significant, association was observed between O_2_^•−^ generated by NOX2 in left atrial appendages (LAA) and left atrial enlargement in AF patients [160]. The serum level of soluble NOX2-derived peptide increased significantly in patients with persistent/PAF and significantly contributed to the increased production of serum isoprostane [64]. Additionally, urine F2-isoprostanes and isofurans were 20% and 50% higher, respectively, in patients with POAF at the end of the surgery [159]. POAF patients were found to have significantly higher basal myocardial O_2_^•−^, NADPH-stimulated O_2_^•−^, and ONOO^−^ production, as reported by Casadei and colleagues [161]. 

Studies on animals have revealed that Rac1-dependent NOX-derived ROS plays a role in AF pathogenesis [56,162,163]. Clinical studies have also established a correlation between AF, NOX2 upregulation, and oxidative stress [64,66,164]. Inhibition of Rac1 and NOX2 activity was observed in right atrial samples of POAF patients treated with atorvastatin. However, atorvastatin did not affect nitric oxide synthase uncoupling, tetrahydrobiopterin levels, or ROS in patients with permanent AF [70]. This suggests that the activation of atrial NOXs is an early and transient event in the natural history of AF. It is possible that changes in ROS sources during atrial remodeling could explain why statins are effective in preventing AF, but not in treating it. In diet-induced obese mice, AF was induced by pacing in all cases, while only 25% of controls exhibited AF [165]. In these mice, cardiac *I_Na_* expression, and atrial APD were decreased while Kv1.5 potassium channel expression and corresponding current (*I_Kur_*), F2-isoprostanes, NOX2, and PKC-a/d expression, and atrial fibrosis significantly increased. A mitochondrial antioxidant restored *I_Na_*, *I_Ca,L_*, *I_Kur_*, APD, reversed atrial fibrosis, and attenuated AF burden in diet-induced obese mice. 

Oxidative stress in mitochondria contributes significantly to cardiac dysfunction in the pathogenesis of POAF [166,167]. In a prospective study, Montaigne et al. examined mitochondrial respiration and myocardial oxidative stress levels in the right atrial tissues of patients with metabolic disease before cardiac surgery [166]. Those with reduced mitochondrial respiration, high sensitivity to calcium-induced mPTP opening, increased ROS production, and upregulation of MnSOD and catalase had a 44% incidence of POAF. Case-control studies have supported this finding, showing lower oxidative phosphorylation and altered gene expression in cardiac mitochondrial energy production in human atrial biopsy specimens from patients with chronic AF [168,169]. NOX4 is a major source of mitochondrial ROS in cardiomyocytes and vascular cells [94,98,102]. Although mitochondrial oxidases are not a major source of ROS in POAF, NOX-derived ROS can damage mitochondrial DNA and protein complexes, leading to mitochondrial dysfunction and, ultimately, cardiac arrhythmia [166,170,171,172]. 

### 3.2. Role of NADPH Oxidases and Oxidative Stress in Permanent AF

Patients with persistent AF experience high atrial activity, leading to excessive energy consumption by cardiac myocytes to maintain excitation-contraction coupling, which can affect their structure and function. Mitochondria, the primary energy source, are also subject to metabolic stress. In individuals with AF, fragmented mitochondria are present in the atrial tissue. When HL-1 atrial myocytes are subjected to tachypacing, mitochondrial Ca^2+^ handling is impaired, mitochondrial membrane potential decreases, and ATP production is reduced [136]. Increased atrial activity alters the mitochondrial redox balance as well. For instance, a study by Yoo and colleagues found that ROS production increases in isolated atrial swine myocytes in a frequency-dependent manner [73]. Additionally, they examined the sources of ROS in atrial samples from dogs subjected to tachypacing and found that mitochondria and NOX2 were significant sources of ROS. Mitochondrial DNA is highly vulnerable to ROS damage, as evidenced by 4977-base deletions of mtDNA and elevated mitochondrial 8-OHdG levels in atrial samples from patients with permanent AF, indicating oxidative stress-induced DNA damage [137]. In a sheep model of persistent AF, Rac1 expression and NOX-stimulated O2^•−^ production increased, while eNOS expression and circulating NO levels decreased [173].

Marks and colleagues showed that mice with RyR2 mutations that cause intracellular Ca^2+^ leak have an increased susceptibility to AF due to mitochondrial dysfunction, ROS production, and atrial RyR2 oxidation [174]. In patients with chronic AF, RyR2 was found to be oxidized, phosphorylated, and depleted of calstabin 2, which plays a role in stabilizing the closed state of RyR2 during diastole [174,175]. Genetic variants in *SCN5A* have also been linked to AF in patients with congenital long QT syndrome and lone AF. Atrial myocytes from chronic AF patients showed a significant reduction in peak *I*_Na_ density and lower expression of Nav1.5, along with an increase in late *I*_Na_ [176,177,178]. By inhibiting *I*_NaL_- *I*_Na_ peak ratio, ranolazine was found to reverse proarrhythmic activity in atrial myocytes and improve diastolic function. Studies by Avula et al. reveal that expressing human mitochondrial catalase in mice overexpressing human F1759A NaV1.5 channels can reduce cardiac structural remodeling, spontaneous AF incidences, and pacing-induced after-depolarizations despite the heterogeneously prolonged atrial action potential [179]. 

Transcriptional oxidative stress response affects permanent AF pathophysiology. In their study, Kim et al. used cDNA microarrays to compare transcription levels of putative genes that might be involved in the oxidative stress response between control patients and patients with chronic atrial fibrillation AF [180]. Results showed that eight genes associated with oxidative stress were upregulated in atrial samples from AF patients, including NOX, while six antioxidant genes were downregulated, such as those encoding glutathione peroxidase, glutathione reductase, and superoxide dismutase. Patients with persistent AF also had higher levels of oxidized glutathione and cysteine in their serum and plasma than those with normal sinus rhythm. Interestingly, EhGSH levels above the median were a stronger predictor of chronic AF than paroxysmal AF [181,182]. In contrast to oxidative stress, no significant correlation was found between proinflammatory markers and AF [182]. 

### 3.3. ROS in the Transition from Paroxysmal to Permanent AF

Paroxysmal AF begins with short episodes that become persistent and eventually permanent over time. As AF advances from short episodes to a persistent state, the atria undergo progressive structural and functional changes that ensure its long-term persistence [183,184]. In the initial stages, electrical remodeling results in the shortening of atrial refractoriness [185,186]. It is unclear how these changes stabilize atrial fibrillation despite known ion channel alterations in animal models and humans [187,188,189]. Structural remodeling and fibrosis result in intra-atrial conduction disturbances. However, their role in the progression of atrial fibrillation from paroxysmal to persistent remains unclear [183,186].

It is undeniable that oxidative stress plays a significant role in the development of AF. However, as AF progresses from paroxysmal to permanent, the source of ROS also changes. According to Riley et al., the mechanism responsible for NO-redox imbalance differs between atria, as well as the duration and substrate of AF [71]. After two weeks of AF, left atrial O_2_^•−^ production increases due to increased NOX expression and activity. Conversely, in cases where AF or atrioventricular block has been persistent, mitochondrial oxidases and uncoupled NOS activity in the right atrium are responsible for the biatrial increase in increase in O_2_^•−^ production (due to a reduction in BH_4_ and/or an increase in arginase activity). It was found that Rac1 activity in RA samples from persistent AF patients was not increased. In addition, atorvastatin did not reduce O_2_^•−^ production in a mevalonate-reversible manner, which indicates that statins may not be beneficial in secondary AF prevention. The group discovered that miR-31 upregulation in goats and patients with persistent AF depletes nNOS and reduces NOS activity due to mRNA decay and the translational repression of dystrophin, leading to the loss of nNOS in the sarcolemmal region [190]. nNOS is constitutively expressed in cardiomyocyte sarcoplasmic reticulum and sarcolemmal membranes as part of the dystrophin-associated glycoprotein complex [191]. nNOS-derived NO controls sarcolemmal ion conductance [192,193] and calcium fluxes [194] and prevents arrhythmic death in mice after myocardial infarction [195]. Upregulation of miR-31 and disruption of nNOS signaling contribute to the electrical remodeling of atrial myocardium in mice and significantly increase AF inducibility. This is done by shortening APD and abolishing rate-dependent adaptation.

Martins et al. studied persistent long-standing AF in sheep and discovered that the rate at which dominant frequency (DF) increases can predict when AF stabilizes and becomes persistent [183]. Transcriptome and proteomic analyses revealed changes in extracellular matrix remodeling, inflammation, ion channels, myofibril structure, and mitochondrial function during the early stages of AF, but not in later stages in sheep [184,196]. CMs in AF have decreased expression in several potassium channel genes (*KCNJ3*, *KCNJ5*), calcium channel genes (*CACNA1C*), or sodium channel genes (*SCN5A*), most notably when comparing control and transition samples. In contrast, others, such as *HCN2* or *KCNH7*, have increased expression [184]. Some of the genes that are upregulated in AF, including *RCAN1* and *LGALS3BP*, have a significant effect on its pathophysiology. Patients with persistent AF show significant DNA damage correlated with poly(ADP)-ribose polymerase activation, NAD^+^ depletion, and oxidative stress in CMs [197]. In patients with permanent AF, about 60% of atrial myocytes are dystrophic, with extensive sarcomere loss and DNA cleavage [196]. These dystrophic myocytes express low levels of antiapoptotic death protein BCL-2, indicating increased susceptibility to death signals compared to control cells.

## 4. Oxidative Stress and Arrhythmogenic Mechanisms in AF

### 4.1. Oxidative Stress Is Associated with AF-Triggered Activity

It has been observed that ectopic beats originating in pulmonary veins can lead to frequent AF paroxysms [8]. By modifying pulmonary vein and left atrial electrophysiological properties, oxidative stress can facilitate AF (Figure 2). Isolated rabbit pulmonary veins (PVs) exposed to H_2_O_2_ (2 mM) demonstrated induced ectopic firing and EADs, as well as shortened APD in PVs and the left atrium [198]. Pretreatment with ascorbic acid (1 mM) or N-(2-mercaptopropionyl)glycine (N-MPG, 10 mM) attenuated H_2_O_2_-induced PV burst firing and increased spontaneous rates. 

Excessive prolongation of the AP can result in EADs, which are caused by the upregulation of *I*_NaL_ and *I*_Ca,L_ currents. This occurrence is linked to oxidative stress and is the primary arrhythmogenic mechanism. In an isolated aged rat atria study, acute oxidative stress through H_2_O_2_ perfusion initiated spontaneous AF that was preceded by EADs in the left atrial epicardial appendages [199]. Direct inhibition of oxidative activation of CaMKII activity or specific blocking of CaMKII-mediated *I*_NaL_ upregulation prevented and suppressed EAD-mediated triggered activity and spontaneous AF (Figure 2). Both atrial tissue fibrosis and acute oxidative stress are necessary for AF initiation, and neither alone is sufficient. In hiPSC-CMs, high epinephrine concentrations, such as those seen in Takotsubo syndrome, decreased *I*_Na_ and enhanced *I*_Ca,L_, prolonging AP duration and inducing arrhythmic events, including EADs and DADs via alpha-1-adrenoceptor stimulation [200]. In this study, alpha-1-adrenoceptor activity involved NOX and protein kinase C activation. In iPSC-CMs, epinephrine and two different D1/D5 receptor agonists caused arrhythmia by reducing depolarization velocity and prolonging APDs [201]. Dopamine receptor signaling was NOX-dependent, and abnormal APs resulted from reduced *I*_Na_ and *I*_Kr_ and upregulation of LTCC channels. 

Human atrial myocytes express Kv1.5 channels that enable ultra-rapid potassium current (*I*_Kur_) and accelerate membrane repolarization during phases 1 and 2 of the AP [202]. Reduction of *I_Kur_* is crucial for the excitability of atrial muscles as seen in patients with a Kv1.5 (E375X) loss-of-function variant who experience idiopathic AF [203]. Blockade of Kv1.5 with 4-aminopyridine (4-AP) prolongs APDs, facilitates EADs in human atrial myocytes, and increases the likelihood of stress-provoked trigger activity. A murine model confirmed the connection between Kv1.5 function and AF susceptibility. Kv1.5 is highly sensitive to redox modification, including S-nitrosylation and S-sulfenylation [204,205]. Sulfenic acid modification of the COOH-terminal C581 in Kv1.5, causes internalization and degradation [205]. In patients with chronic AF, there was a global increase in sulfenic acid-modified proteins and sulfenic acid modification of Kv1.5, indicating the presence of Kv1.5 dysfunction in AF.

### 4.2. ROS Alter the Intracellular Ca^2+^ Handling

Excitation-contraction (E-C) coupling connects the electrical activation of cardiac muscle cells to their mechanical contraction [206]. During this process, membrane depolarization causes Ca^2+^ to enter through L-type Ca^2+^ channels, triggering the release of Ca^2+^ from the SR, resulting in myocyte contraction [207]. During relaxation, RyR2 and LTCC become nonconductive, while SERCA ATPase and NCX pump Ca^2+^ back into the SR and out of the cell, respectively (Figure 3). The characteristics of Ca^2+^ flux are determined by the levels of intracellular Ca^2+^ receptors/channels, just as specific ion channels expressed in atrial cell membranes determine the triangular AP waveform [207,208]. For example, in atrial myocytes, Ca^2+^ transient amplitudes are smaller, and intracellular Ca^2+^ decay is faster due to an increased expression of SERCA and a lower expression of SERCA inhibitory protein phospholamban (PLB). The increased function of SERCA results in a higher Ca^2+^ content in atrial myocytes than ventricular myocytes. The greater atrial SR Ca^2+^ content makes atrial myocytes more susceptible to spontaneous diastolic SR Ca^2+^ release when RyR2 channels are sensitized under pathological conditions [209,210,211,212]. At a structural level, atrial myocytes lack well-developed T-tubules, which means that Ca^2+^ signals start at the cell’s outer edge during E-C coupling [207,213]. Overall, the electrophysiological, structural, and Ca^2+^ flux properties of the atrial myocytes play a role in their efficient E-C coupling and their ability to sustain high excitation frequencies in AF. 

SR membranes contain NOX, supporting the notion that oxidative stress plays a significant role in the Ca^2+^ handling properties of myocytes [214]. Confocal experiments have shown that NOX colocalizes with RyR2, suggesting that ROS produced by NOX regulate SR Ca^2+^ release [215]. It is worth noting that each RyR2 monomer contains at least 21 cysteine residues that are susceptible to oxidation [216]. Hydroxyl radicals and SH oxidizing agents can enhance the open probability of RyR2 channels reconstituted in lipid bilayers, leading to an increase in Ca^2+^ release from the SR. At the same time, dithiothreitol reverses the effects of SH oxidizing agents [217,218]. Oxidizing agents also increase RyR2-mediated Ca^2+^ leak in permeabilized isolated ventricular myocytes, further supporting the role of redox regulation of RyR2 in heart failure [219]. Isolated SR vesicles have been used to show that NOX activity, RyR2 S-glutathionylation, and calcium release rates are increased during exercise and tachycardia [220]. 

Cardiomyocytes express all three NOS isoforms [221]. Gonzales et al. [222] found that NOS1 deficiency causes RyR2 hyponitrosylation and enhanced channel oxidation, resulting in diastolic Ca^2+^ leak and a proarrhythmic phenotype in mice. RyR2 oxidation can lead to inhibition instead of activation. When SR membrane preparations were treated with a physiological concentration of NADH-activated NOX, Ca^2+^-induced Ca^2+^ release (CICR) was inhibited, with RyR2 single channel open probability significantly reduced [214]. In permeabilized myocytes, activation of NOX reduced the frequency and amplitude of diastolic spontaneous Ca^2+^ release. An increase in RyR2 S-glutathionylation, induced by NOX, resulted in a preconditioning response [215]. The function of RyR2 may be affected differently by oxidative stress due to variations in ROS production, type, and duration of exposure. RyR2 oxidation resulted in concentration and time-dependent changes in the activation threshold for store overload-induced Ca^2+^ release (SOICR) [223]. Low levels of RyR2 oxidation increased channel activity by decreasing the threshold for SOICR, whereas high levels irreversibly increased the threshold for SOICR, leading to RyR2 inhibition.

In patients with chronic AF, impaired diastolic closure of SR RyR2 is associated with increased serine 2808 phosphorylation of RyR2 and decreased binding levels of FKBP12.6 (calstabin 2), which is an inhibitory subunit of RyR2 [224,225]. Atrial myocytes from *FKBP12.6^−/−^* mice have increased diastolic SR Ca^2+^ leak, spontaneous Ca^2+^ release, and greater vulnerability to pacing-induced AF [225]. A mouse model of CPVT (catecholaminergic polymorphic ventricular tachycardia) with highly oxidized RyR2 showed increased SR Ca^2+^ leak and propensity for AF [226]. The oxidation of RyR2 was associated with the depletion of FKBP from RyR2, providing the molecular basis for atrial arrhythmia (Figure 3). Pharmacological stabilization of RyR2-FKBP binding prevented diastolic spontaneous Ca^2+^ release and reduced the likelihood of atrial arrhythmia in isolated atrial myocytes. A disruption of RyR2-FKBP interaction caused by oxidative stress may also contribute to RyR2 leak and cardiac arrhythmia caused by palmitoyl-carnitine [227]. Mitochondrial ROS can affect RyR2-FKBP stability, and the changes in RyR2 lead to heightened sensitivity to Ca^2+^-induced activation at low cytosolic Ca^2+^ levels. This increases spontaneous Ca^2+^ release during diastole [224,228]. PLB phosphorylation may further enhance the activity of SERCA2a and increase the likelihood of SR Ca^2+^ releases in atrial myocytes from AF patients [229,230]. During diastole, the rise in cytosolic Ca^2+^ concentrations can lead to an inward Na^+^/Ca^2+^ exchange current, resulting in DADs, which may cause atrial arrhythmias [229,231].

In conditions of increased ROS, the activity of CaMKII (Calcium/calmodulin (Ca^2+^/CaM)-dependent protein kinase II) is more responsive to an elevation of Ca^2+^ [232] (Figure 3). Anderson and colleagues discovered higher levels of oxidized CaMKII in the atrial tissues of patients with AF. They also showed the essential role of CaMKII in ANGII/pacing-induced AF using mouse models [123,233]. Infusing Ang II for three weeks increases ox-CaMKII levels in atrial tissue [80], which causes SR Ca^2+^ leak in isolated atrial myocytes and makes mice more susceptible to AF induction. Mice overexpressing methionine sulfoxide reductase A, an enzyme that reduces oxidized CaMKII, and mice with oxidation resistant CaMKII were resistant to AF induction following Ang II infusion, suggesting CaMKII is a mechanistic link between oxidative stress and AF [234]. In mice treated with ibrutinib, a drug that increases the risk of AF development in cancer patients, there was an increase in ox-CaMKII, p-CaMKII (Thr-286), and p-RyR2 (Ser2814) expression, as well as increased SR Ca^2+^ leak, abnormal mitochondrial structures in atrial cardiomyocytes, atrial fibrosis, and pacing-induced AF [235,236]. Additionally, there was increased NOX2, NOX4, xanthine oxidase, and TGF-β1 protein expression in atrial tissue and mitochondrial ROS levels in cardiomyocytes. Apocynin decreased ibrutinib-induced oxidative stress protein expression and propensity to AF in mice [235].

The SERCA pump serves two purposes—to reduce cytosolic Ca^2+^ levels to relax muscles and to restore the Ca^2+^ load necessary for muscle contraction [237]. Proper regulation of the SR Ca^2+^ release is crucial for maintaining mitochondrial calcium homeostasis [238]. SERCA Cys674 is a critical target for oxidative (sulfonation) modification [239,240]. Like RyR2, SERCA can either be activated or inhibited by oxidation, depending on the type of ROS involved, the duration of exposure, and the concentration. Short-term exposure to H_2_O_2_ leads to a decrease in SR Ca^2+^ content, Ca^2+^ amplitude, and cell shortening before diastolic arrest in ventricular myocytes [241]. Additionally, SR membrane preparations exposed to H_2_O_2_ show reduced Ca^2+^ recapture, which is consistent with the inhibitory effects of ROS on SERCA activity [238]. Loss of SERCA function due to oxidative stress is responsible for diastolic dysfunction in aging, acquired, or inherited cardiac conditions, as well as metabolic dysregulation [242,243,244,245]. In contrast, exposure of ventricular myocytes with 500 mol/L Nitroxyl (HNO) for 15 min increased the maximal rate of thapsigargin-sensitive Ca^2+^ uptake via reversible oxidation of SERCA thiol groups [246]. This was correlated with increased levels of S-glutathionylation at cysteine 674. Cardiomyocytes expressing a redox-insensible mutant of C674 (C674S) showed decreased SR Ca^2+^ content, consistent with an upregulation of SERCA by oxidation. SERCA oxidation was also associated with a higher mitochondrial Ca^2+^ concentration. Lastly, heterozygous knock-in of SERCAC674S in mice with chronic ascending aortic constriction attenuated myocyte apoptosis, LV dilation, and systolic dysfunction. Further investigations are required to determine whether SERCA oxidation is a dominant factor in AF genesis.

### 4.3. Oxidative Stress Is Associated with Functional Reentry

Functional reentry occurs when an impulse propagates around a refractory core without any underlying substrate or anatomical barriers [247]. This is made possible by altering the function and expression of membrane ion channels, shortening the AP and refractory periods [248]. In the early stages of AF, the reduction of *I*_CaL_ occurs as a response to the Ca^2+^ overload imposed by the high atrial excitability, which results in Ca^2+^-dependent inactivation. As the arrhythmia progresses, *I*_Ca,L_ reduction continues through downregulating the α-subunit of the LTCC (Cav_1_._2_) [249]. Many clinical and experimental studies have shown that oxidative stress plays a significant role in reducing *I*_Ca,L_ levels. Carnes and colleagues [250] discovered that N-acetylcysteine can help restore the function of the *I*_Ca,L_ in atrial myocytes isolated from AF patients. In line with this, acute exposure of cardiomyocytes from the left atria to H_2_O_2_ led to a reduction *I*_Ca,L_, and a shorter APD [251]. Therefore, oxidative stress not only causes EADs, as mentioned earlier, but also sustains AF by causing functional reentry. In an in vitro model of AF, 3-morpholino-sydnonimine (SIN-1), a NO donor, prevented the shortening of the atrial refractory period and restored the atrial rhythm. Direct exposure of atrial myocytes to SIN-1 increased *I*_Ca,L_, highlighting the significance of LTCC downregulation in AF pathogenesis due to imbalances in ROS/NOS production [173].

In AF, the downregulation of *I*_Na_ is an ionic mechanism that promotes reentry by impairing conduction [186]. Several studies have examined the effect of ROS on peak *I*_Na_. Rat neonatal cardiomyocytes treated with NADH, a substrate for NOX, show a rapid decrease in *I*_Na_, which can be prevented by superoxide dismutase [252]. It is suggested that oxidative stress may promote reentry, as perfusion of NADH into the rabbit heart increases the propensity to AF [253]. Oxidative stress also affects *I*_Na_ by modulating the expression of the major in the heart, Na_V_1.5 encoded by *SCN5A*. Ang II and H_2_O_2_ promote the binding of NF-kB to the cardiac *Scn5a* promoter, reducing *Scn5a* mRNA levels [254]. FoxO transcription factors, known for regulating cellular processes including aging, play a significant role in regulating *Scn5a* transcription [255]. Studies predict FoxO binding elements in *Scn5a* promoter, and FoxO deletion upregulates *Scn5a* transcription [256]. FoxO1 inhibits Nav1.5 expression by binding to the proximal *Scn5a* promoter region [257]. Upregulation of Nav1.5 expression requires the translocation of FoxO1 from the nucleus to the cytosol. FoxO1 localization is sensitive to oxidative stress. In HL-1 atrial cardiomyocytes, H_2_O_2_ exposure reduces Nav1.5 channel expression and *I*_Na_ levels by retaining FoxO1 in the nucleus.

The inwardly rectifying potassium current *I*_K1_ largely determines the resting membrane potential and the terminal repolarization of the cardiac action potential [258]. Specifically, cardiac *I*_K1_ is mediated by Kir2.x channels [259]. In chronic AF patients, increased expression of Kir2.1 expression and *I*_K1_ can lead to electrical remodeling, which contributes to the maintenance of arrhythmia by reducing the refractory period [260,261]. Moreover, the differential Kir2.1 expression in the right and left atrium can contribute to high-frequency entry sources in paroxysmal AF [188].

### 4.4. ROS Promote Anatomical Reentry Mechanisms

Gap junctions are clusters of channels that connect adjacent cells for chemical and electrical communication. Atrial gap junctions comprise connexin proteins Cx40 and Cx43 [262]. Changes in the expression and location of connexins can cause AF by slowing heart conduction, creating unidirectional blockages, and forming reentrant circuits [263,264] (Figure 4). Connexins also impact the heterogeneity of refractoriness, which is another significant factor in reentry. When cells are tightly connected, it normalizes variations in APD [265]. During several pathological conditions that increase the risk of AF, connexins are downregulated, and recent research shows that oxidative modification may play a role. The end binding 1 (EB1) protein binds to microtubules and helps transfer Cx43 to adherens junctions [266]. When exposed to oxidative stress, EB1 dissociates from microtubules in human end-stage ischemic cardiomyopathy, the ischemia-reperfusion model in isolated mouse hearts, and cell lines exposed to H_2_O_2_. This reduces the forward trafficking of Cx43 [267]. Moreover, this study also found that H_2_O_2_ exposure slowed cardiac excitation and decreased cardiac gap junction coupling in zebrafish. ROS can also directly target Cx43 and alter its distribution. In a mouse model of diabetes, there were increased levels of oxidized connexin 43 tyrosine sites, leading to Cx43 mislocalization and abnormal ventricular conduction patterns [268]. In a sleep apnea mouse model, the expression of Cx40 and Cx43 in atrial tissue and the size of the gap junctions were reduced [70] (Figure 4). Interestingly, NOX2-deficient mice do not exhibit these effects, indicating that oxidative stress effects on connexin expression are NOX2-dependent. These findings demonstrate how oxidative stress regulates the underlying mechanisms of sleep apnea predisposes to AF.

Augmented fibrosis in the atria can cause reentry mechanisms by impacting conduction velocity [18,269]. Fibrosis can also lead to a voltage gradient, known as the “source-to-sink effect,” between neighboring cardiomyocytes, resulting in reentry. The simultaneous synthesis and degradation of extracellular matrix components determines the degree and location of cardiac fibrosis. Antioxidant studies have shown that oxidative stress is linked to the development of fibrosis in cardiac stress conditions. Resveratrol, when administered intraperitoneally, can prevent the development of atrial fibrosis in rabbits and rats, thus decreasing the likelihood of AF in electrical pacing studies. The activation of profibrotic signaling pathways highly depends on NOX activation, suggesting a relationship between oxidative stress and fibrosis. *Nox2* null mice had significantly reduced profibrotic effects following Ang II infusion [270]. Additionally, fibroblasts express NOX4 and NOX5, and their activity, particularly NOX4, is crucial for differentiating fibroblasts into myofibroblasts caused by TGF-β1 [271]. Finally, genetic elimination of the NOX1 and NOX2 regulatory subunit Rac1 prevented fibrosis development in a mouse model of diabetes [101].

The degradation of fibrotic components in the extracellular matrix is facilitated by metalloproteinases (MMPs), resulting in cardiac structural remodeling. According to Nakano et al., the expression and activity of MMPs are changed in fibrillating atria, leading to atrial structural remodeling and dilatation during AF [272]. Activation of MMPs requires proteolytic cleavage of inactive peptides (pro-MMPs). Purified proMMP-1, -8, and -9 are activated by peroxynitrite, as demonstrated by Okamoto and coworkers, indicating that posttranslational modification is also a mechanism for modulating MMP activity [273]. This is supported by the finding that activation of xanthine oxidase in isolated fibroblasts or exposure to H_2_O_2_ enhanced the activation of MMP-1, MMP-2, and MMP-9, and prevented collagen accumulation [274]. Moreover, myocardial structural remodeling is regulated by oxidative stress, as pharmacological inhibition of SOD increases collagen production [274,275]. Moe and coworkers provided direct evidence of MMP involvement in AF pathogenesis by showing that inhibiting MMP-1 prevented atrial fibrosis in a canine heart failure model, and reduced AF severity and duration [276].

In cases of permanent AF, oxidative stress can lead to structural remodeling that impacts the contractile machinery of atrial myocytes. Research has shown that tyrosine nitration and carbonylation of the myofibrils can cause dysfunction, rather than changes in composition [168]. Studies conducted in vivo and ex vivo experiments with tachypacing have suggested that Ca^2+^-activated calpains are responsible for degrading myofibrils in cases of permanent AF [277,278]. This degradation may be further exacerbated by increased intracellular Ca^2+^ levels resulting from the ROS-induced ionotropic effect on LTCC during AF.

A proteomic analysis was conducted on the tissue of LAA from patients with AF and those with sinus rhythm. The results indicated that mitochondrial dysfunction, oxidative phosphorylation, and Sirtuin signaling were the most affected pathways [279]. An increase in the expression of electron transport chain subunit proteins was observed, while several assembly factors decreased. The upregulation of the plasma membrane lipid transporter CD36 was not accompanied by an upregulation of proteins facilitating fatty acid uptake or utilization. Instead, the expression of key enzymes in fatty acid utilization, such as ACSS2, ACOT, and ACACB, decreased. Several glycolytic enzymes, including ENO3, GPD2, and GAPDH, also decreased. These findings suggest that, while the LAA tries to meet the energetic demands of AF, an uncoordinated response may lead to a decrease in ATP availability. This, in turn, could contribute to tissue contractile and electrophysiologic heterogeneity, and promote the progression of AF from paroxysmal episodes to the development of a substrate amenable to persistent arrhythmias.

The enlargement of the atrium is an independent risk factor for the development of AF [280,281]. This is because the increase in space allows for the formation of reentry circuits [282]. Patients with sustained AF often have elevated plasma levels of oxidative markers and atrial enlargement [283,284,285]. Experimental studies have shown that oxidative stress is causal in ventricular hypertrophy [286,287,288]. Increased NOX2 expression and activity are the major sources of ROS generation in pressure overload left ventricular hypertrophy [289,290]. Hypertrophic myocardium with impaired diastolic relaxation causes elevated left atrial pressure, leading to atrial stress and enlargement, which can lead to the development of AF [291,292]. Chronic atrial stretch activates many signaling pathways, resulting in cellular hypertrophy, fibroblast proliferation, and tissue fibrosis [281]. Dilated patients have an increased non-uniform anisotropy and a macroscopic slowing of conduction in their electroanatomical substrate, promoting the formation of reentrant circuits.

## 5. Concluding Remarks

The electrical and structural remodeling of the atria, as well as the initiation and progression of paroxysmal to permanent AF, are significantly impacted by oxidative stress from NOX activation and mitochondrial dysfunction. Mitochondrial dysfunction can be caused by NOX-derived ROS, which can damage mitochondrial DNA and protein complexes, leading to AF associated with aging. Uncoordinated responses to meet the energetic demands of AF can reduce ATP availability, negatively impacting calcium signaling and the contractile machinery of cardiomyocytes. This dysregulation can create heterogeneity in atrial contractility and electrophysiology, leading to persistent arrhythmia. Patients with permanent AF, heart failure, and sinus rhythm typically have dystrophic atrial myocytes with extensive sarcomere loss and DNA damage. Changes in extracellular matrix remodeling, inflammation, ion channels, myofibril structure, and mitochondrial function occur during the early transitional stages of AF but not later. Given the limited effectiveness of antiarrhythmic drugs for AF, new treatment options for pAF patients and inhibiting the transition from paroxysmal AF to permanent AF include isoform-specific NOX inhibitors, mitochondrial ROS scavengers, and drugs that target mitochondrial dynamics and metabolism.

## Figures and Tables

**Figure 1 antioxidants-12-01833-f001:**
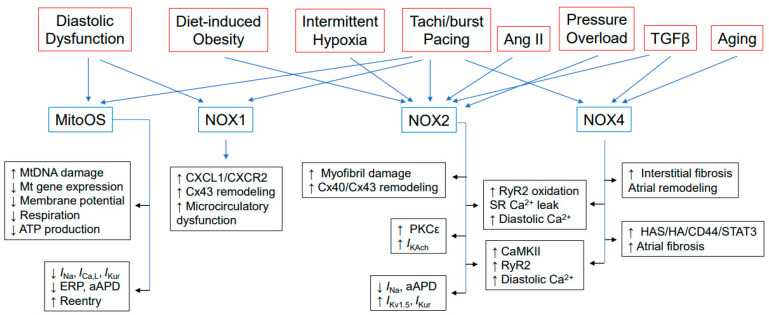
Oxidative stress from dysregulated NOX activation and mitochondrial dysfunction regulates the signaling pathways involved in initiating and maintaining AF. The hyperactivation of NOX1, NOX2, and NOX4 can regulate various signaling pathways in the heart that are involved in the initiation and maintenance of AF. When NOX1 levels/activity are increased, it can lead to microvascular dysfunction in diastolic dysfunction and metabolic syndrome, risk factors for AF. Excess NOX2 activity triggers the onset of AF and is linked to myofibril damage and decreased connexin expression. Additionally, NOX2 can activate PKCε and increase the constitutively active form of acetylcholine-dependent K^+^ current (*I_KAch_*). Increased NOX2 expression is associated with reduced cardiac channel expression (*I_Na_*) and atrial APD in pacing-induced AF in diet-induced obese mice. Potassium channel expression (Kv1.5) and current (*I_Kur_*) were significantly increased. Baseline serum NOX4 levels in patients with AF have a significant independent association with paroxysmal and persistent/permanent AF. RyR2 oxidation, diastolic SR Ca^2+^ leak, and increased NOX2 and NOX4 expression are causatively associated with AF. NOX4 can also induce CaMKII activation, which increases susceptibility to AF. Tachypacing-induced Ca^+^-handling abnormalities in HL-1 myocytes are mediated by NOX2/NOX4-induced HAS/HA/CD44 signaling pathway. NOX4 and CD44 increase atrial fibrosis and susceptibility to AF. Cardiomyocytes from patients with AF exhibit increased levels of ROS in the mitochondria, structural abnormalities, impaired function, decreased ATP production, and reduced expression of enzymes involved in fatty acid utilization and glycolysis. These mitochondrial dysfunctions and myocardial oxidative stress contribute to shortened APDs, which lead to reentry.

**Figure 2 antioxidants-12-01833-f002:**
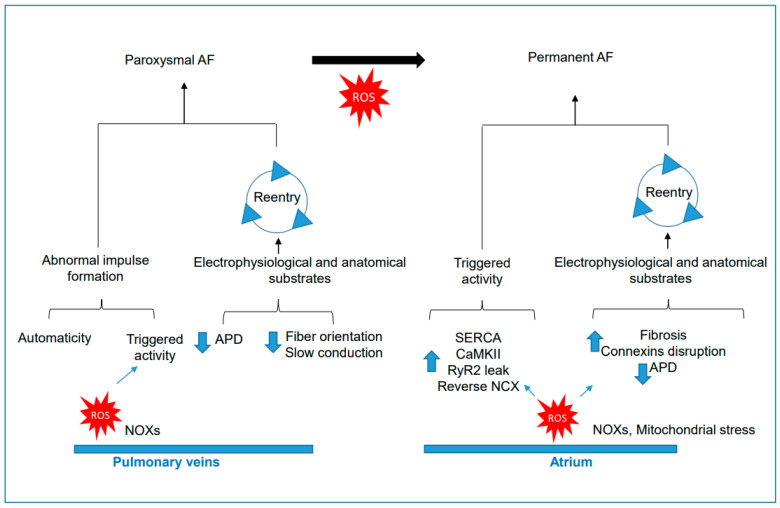
Oxidative stress is a critical factor in the onset and progression of AF. The susceptibility to paroxysmal AF is increased due to automaticity, triggered activity, and electrophysiological and structural remodeling of pulmonary veins. The arrhythmia triggers are promoted by ROS, which lead to afterdepolarizations. Electrophysiological and anatomical alterations in pulmonary veins cause conduction slowing and altered depolarizing and repolarizing currents, which, in turn, promote reentry-triggered activity. ROS target sarcoplasmic reticulum proteins, causing RyR2 dysfunction and increased Ca^2+^ leak, activation of NXC in reverse mode, and triggered activity in atrial CMs. Additionally, ROS contribute to AF progression by inducing electrophysiological and structural remodeling in atria. Membrane current flux is altered by ROS, accelerating membrane repolarization and shortening the refractory period, thereby accelerating reentry. In structural remodeling, ROS enhance profibrotic pathways and disarray integral components of gap junctions (connexins), slowing cardiac conduction or increasing conduction heterogeneity, favoring reentry.

**Figure 3 antioxidants-12-01833-f003:**
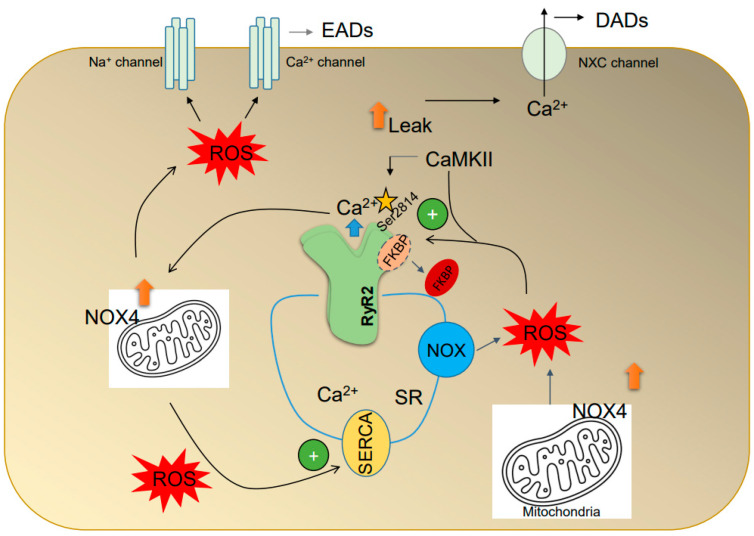
ROS regulate Ca^2+^ fluxes in atrial cardiomyocytes. NOXs located in the sarcoplasmic reticulum (SR) or mitochondria regulate the ryanodine receptor 2 (RyR2) and the sarco-endoplasmic reticulum calcium ATPase (SERCA) pump. Oxidation of RyR2 causes FKBP dissociation, which makes the channel leaky. Furthermore, oxidative activation of CaMKII phosphorylates the RyR2 channels at Ser2814, causing the channel to leak. The increase in calcium levels activates the forward mode of the sodium-calcium exchanger (NCX), which results in delayed afterdepolarizations (DADs). Moreover, ROS can dysregulate the function of sodium and calcium channels, causing slower membrane repolarization and early afterdepolarizations (EADs), which may lead to AF. Thus, ROS play a crucial role in maintaining Ca_2_^+^ flux between the SR and mitochondria.

**Figure 4 antioxidants-12-01833-f004:**
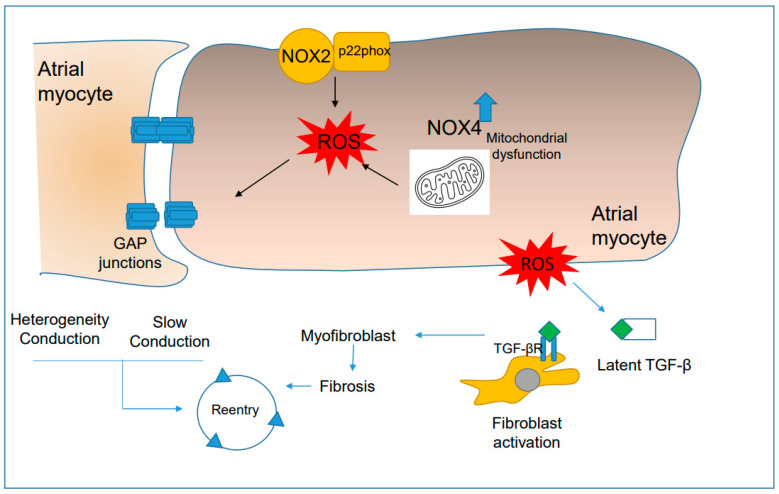
ROS promote cardiac structural remodeling. Increased ROS production promotes cardiac remodeling by targeting connexin integrity in CMs, impairing conduction velocity and promoting conduction heterogeneity. ROS activate TGF-β, which induces fibroblast differentiation, leading to fibrosis deposition and anisotropic cardiac conduction.

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
