# Peer review of "NADPH Oxidases and Oxidative Stress in the Pathogenesis of Atrial Fibrillation"

_antioxidants, 2023, doi:10.3390/antiox12101833_

Round 1

Reviewer 1 Report

 Authors Ramos-Mondrago′n et al. presented a narrative review paper on the involvement of oxidative stress from NOX activation and mitochondrial dysfunction in the electrical and structural remodeling of the atria and the progression of atrial fibrillation. The review is well-written, and the amount of cited literature is adequate.

Before publication, I suggest checking for typos (i.e., sometimes in TGF-b1 the b is missing). Please check also the references format as suggested in Instructions for Authors.

Author Response

Authors Ramos-Mondrago′n et al. presented a narrative review paper on the involvement of oxidative stress from NOX activation and mitochondrial dysfunction in the electrical and structural remodeling of the atria and the progression of atrial fibrillation. The review is well-written, and the amount of cited literature is adequate.

The reviewer's positive comments are greatly appreciated.

Comments:

•  Before publication, I suggest checking for typos (i.e., sometimes in TGF-b1 the b is missing). Please check also the references format as suggested in Instructions for Authors.

The manuscript underwent meticulous English revision and typo correction. Additionally, references were formatted according to the Instructions for Authors.

Reviewer 2 Report

This is an extraordinary well-written and comprehensive treatment of the subject. It will be of great value to all who are studying this arrhythmia, and also to those who are not but who wish to review cellular signalling mechanisms and cellular mechanisms involved in the pathophysiology of cardiac arrhythmias and chamber dysfunction. The thorough treatment of the role of oxidative stress makes surprisingly interesting reading!

I have a small number of suggestions and corrections for the authors to consider.

 From an epidemiologic prospective, and while acknowledging issues of uncertainty, the review might too often impute cause when what has to date been identified is in fact only association.

Abstract - AF is the most common cardiac arrhythmia with clinical significance. It is not the frequency or irregularity of atrial contractions that is of significance in the formation of local clots, but rather their ineffectiveness in displacing blood, leading to local stasis.

Line 77 page 2, intracellular space?

Line 83 page 2, delete "patients" and "in animals"

Line 93 page 2, rephrase "slow conduction time" - reduce time or speed?

Line 115 page 3, rephrase

Line 153 page 4, "DD" - use of nomenclature and short forms is generally excellent, but there are a few such as this that are not included in the text or list of abbreviations.

Line 161 page 4, change "hypersensitive" to "hypertensive"

Line 169 page 4, "bioavailability" - this doesn't seem to be the best word even though it is used in the original reference - "availability" might be a less misleading term

Figure 1, "Pressure Overload" is not identified in the diagram as playing any specific role

Line 199 page 5, "intermittent hypoxia"

Line 219 page 5, "and increased magnitude"

Line 224 page 5, "and reduction in NOX2 activity"

Line 329, "higher levels of HA..."

Line 338, "oxidative"

Line 347, "potential factor in AF pathogenesis…"

Line 416, "have supported these data" or "have supported this finding"

Line 532, there is something wrong with this sentence?

Line 547. This is an example of the issue of cause versus effect. Should the end of the sentence not say "… Indicating the presence of Kv1.5 dysfunction in AF", or you could even leave this out.

Line 555, "ROS targets sarcoplasmic reticulum…"

Line 604. The sentence starting on this line appears to be incomplete.

Line 884. There are spontaneous font changes in the bibliography and spacing issues. 

Author Response

This is an extraordinary well-written and comprehensive treatment of the subject. It will be of great value to all who are studying this arrhythmia, and also to those who are not but who wish to review cellular signalling mechanisms and cellular mechanisms involved in the pathophysiology of cardiac arrhythmias and chamber dysfunction. The thorough treatment of the role of oxidative stress makes surprisingly interesting read.

We appreciate the reviewer for their positive feedback and helpful suggestions.

  • From an epidemiologic prospective, and while acknowledging issues of uncertainty, the review might too often impute cause when what has to date been identified is in fact only association.

We acknowledge that the mechanisms related to AF are more of an association rather than a cause, as the word "associated" appears 40 times in the review.

  • Abstract - AF is the most common cardiac arrhythmia with clinical significance. It is not the frequency or irregularity of atrial contractions that is of significance in the formation of local clots, but rather their ineffectiveness in displacing blood, leading to local stasis.

  We agree with the reviewer and have restructured the sentence as follows.

The irregular and rapid contraction of the atria can lead to ineffective blood pumping, local blood stasis, blood clots, ischemic stroke, and heart failure.

  • Line 77 page 2, intracellular space?

         We would like to express our gratitude to the reviewer for bringing this to our attention. As a result, we have made the necessary revisions to the sentence.

In this operating mode, one calcium ion is extruded from the cells in exchange for three sodium ions entering the cell.

  • Line 83 page 2, delete "patients" and "in animals"

    We have made the necessary corrections.

  • Line 93 page 2, rephrase "slow conduction time" - reduce time or speed?

     We've revised it to slow conduction velocity.  Much appreciated.

  • Line 115 page 3, rephrase

  The following revisions have been made.

Cardiovascular cells produce ROS through various enzymes such as NOX, xanthine oxidase, cyclooxygenases, lipoxygenases, myeloperoxidase, uncoupled endothelial NO synthase, and mitochondrial respiratory chain complexes.

  • Line 153 page 4, "DD" - use of nomenclature and short forms is generally excellent, but there are a few such as this that are not included in the text or list of abbreviations.

New-onset AF and early AF recurrence after ablation are associated with left ventricular diastolic dysfunction (DD) [43-49]. Page 4, last sentence.  DD is also included in the list of Abbreviations.

  • Line 161 page 4, change "hypersensitive" to "hypertensive”

      Revised as below.

Spontaneously hypertensive rats (SHRs) express more CXCL1 and CXCR2 mRNA in response to atrial rapid (burst) pacing [55]  Page 5, line 9.

  • Line 169, page 4, "bioavailability" - this doesn't seem to be the best word even though it is used in the original reference - "availability" might be a less misleading term

Revised as recommended.

Studies have shown a decrease in nitric oxide availability in the left atrium of swine with pacing-induced AF, indicating an oxidative stress-mediated degradation mechanism [56].

  • Figure 1, "Pressure Overload" is not identified in the diagram as playing any specific role

The revised Figure shows Pressure overload inducing NOX2 activity.

 Line 199 page 5, "intermittent hypoxia”

      Corrected.

  • Line 219 page 5, "and increased magnitude

Corrected.

  • Line 224 page 5, "and reduction in NOX2 activity

Corrected.

  • Line 329, "higher levels of HA…”

Corrected.

  • Line 338, "oxidative"

Corrected.

  • Line 347, "potential factor in AF pathogenesis…" 

Revised as below.

These findings suggest that CD44/NOX4 signaling could potentially cause irregularities in Ca2+ handling and lead to AF pathogenesis in certain pathophysiological circumstances.

  • Line 416, "have supported these data" or "have supported this finding"

Corrected.

  • Line 532, there is something wrong with this sentence?

      In hiPSC-CMs, high epinephrine concentrations, such as those seen in Takotsubo syndrome, decreased INa and enhanced ICaL, prolonging AP duration and inducing arrhythmic events, including EADs and DADs via alpha-1-adrenoceptor stimulation [202]. Page 11, Line 1.

  • Line 547. This is an example of the issue of cause versus effect. Should the end of the sentence not say "… Indicating the presence of Kv1.5 dysfunction in AF", or you could even leave this out.

In patients with chronic AF, there was a global increase in sulfenic acid-modified proteins and sulfenic acid modification of Kv1.5, indicating the presence of Kv1.5 dysfunction in AF. Page 11, 2nd paragraph, last sentence.

  • Line 555, "ROS targets sarcoplasmic reticulum…"

In the review, we referred to "ROS" in its plural form.

  • Line 604. The sentence starting on this line appears to be incomplete.

Revised as below.

The function of RyR2 may be affected differently by oxidative stress due to variations in ROS production, type, and duration of exposure.

  • Line 884. There are spontaneous font changes in the bibliography and spacing issues.

The references were formatted according to the Instructions for Authors in the revised manuscript.

Reviewer 3 Report

Atrial fibrillation (AF) is a common cardiac arrhythmia characterized by irregular and often rapid electrical activity in the atria of the heart. There are several mechanisms that contribute to the development and maintenance of atrial fibrillation, such as, 1) Electrical remodeling referring to changes in the electrical properties of atrial tissue that promote the development and persistence of AF; 2) Reentry circuits which electrical impulses circulate within the atria due to areas of unidirectional block and slow conduction; 3) Structural remodeling in the atria, such as fibrosis and atrial enlargement; 4) Abnormal automaticity and triggered activity; 5) Autonomic nervous system unbalance; 6) Mutations in genes encoding ion channels or proteins involved in calcium handling can increase susceptibility to AF; 7) Inflammation and oxidative stress can promote structural and electrical remodeling of the atria, contributing to AF. This review was fucus on the NADPH Oxidases and Oxidative Stress in the Pathogenesis of Atrial Fibrillation. This should be an interesting and substantive review article, but the framework/organization of the review is not easily understood by the reader. 

For example, the authors did not clearly explain the relationship between NOX1 and CXC1. It would have been better for them to add some more information such as NOX1 and CXCL1 are not directly linked in biochemical interactions, but they are functionally linked in the context of inflammation and oxidative stress. NOX1-derived ROS can stimulate the expression of pro-inflammatory genes like CXCL1, and the subsequent action of CXCL1 in recruiting immune cells can further exacerbate the inflammatory response.

Also, the Figure 1 needs figure legend as other figures. There are also many typos and grammatical issues, and authors are advised to review the manuscript again carefully for typos, grammatical errors, and redundant sentences.

There are also many typos and grammatical issues, and authors are advised to review the manuscript again carefully for typos, grammatical errors, and redundant sentences.

Author Response

This review focused on the NADPH Oxidases and Oxidative Stress in the Pathogenesis of Atrial Fibrillation. This should be an interesting and substantive review article, but the framework/organization of the review is not easily understood by the reader. 

We appreciate the reviewer's insightful feedback.

  • The authors did not clearly explain the relationship between NOX1 and CXC1. It would have been better for them to add some more information such as NOX1 and CXCL1 are not directly linked in biochemical interactions, but they are functionally linked in the context of inflammation and oxidative stress. NOX1-derived ROS can stimulate the expression of pro-inflammatory genes like CXCL1, and the subsequent action of CXCL1 in recruiting immune cells can further exacerbate the inflammatory response.

We have now included the following on page 5, paragraph 1, Lines 7-9.

NOX1-derived ROS can stimulate the expression of pro-inflammatory genes, including CXCL1. The resulting action of CXCL1 in recruiting immune cells can further worsen the inflammatory response.

  • Figure 1 needs figure legend as other figures.

A comprehensive legend is included in the revised manuscript for Figure 1.

  • The authors are advised to review the manuscript again carefully for typos, grammatical errors, and redundant sentences.

The manuscript underwent meticulous English revision and typo correction.

Reviewer 4 Report

Dear authors,

This is, in my opinion, already a very well thought-out and written manuscript. My only remarks would be:

-) Why did you do what you did? Maybe explain this in more detail at the end of the Introduction.

-) Methods section: This is missing. I would classify your paper as a narrative review (right?), so please add a respective Methods section describing what you did (also have a look at PRISMA etc., and include your search strategy).

-) Overall text: Please introduce more subheadings and make sure the structure of your - long - manuscript is very clear and not confusing for someone just reading this for the first time.

-) Sentence structure: The manuscript is already long, please try to make your sentence structures a bit simpler to make reading easier.

-) Sentence structure: The manuscript is already long, please try to make your sentence structures a bit simpler to make reading easier.

Author Response

This is, in my opinion, already a very well thought-out and written manuscript.

The reviewer's positive comments are greatly appreciated.

  • Please add a respective Methods section describing what you did.

A description of the methodology used in this narrative review can be found in the revised manuscript on page 4, paragraph 3.

  • Please introduce more subheadings and make sure the structure of your - long - manuscript is very clear

To improve readability, we have broken up the long paragraphs and avoided passive voice usage.

  • Please try to make your sentence structures a bit simpler to make reading easier.

Whenever possible, the revised manuscript was written in short sentences to make it easier for readers to comprehend.

Round 2

Reviewer 2 Report

The extensive revisions made by the authors more than satisfy the general concerns that I identified in my previous review. This is now a very valuable contribution to the literature.

Reviewer 3 Report

The manuscript is a significant improvement over the previous version in terms of writing, description of methods, and arrangement of data. And please double-check the manuscript for typos, e.g., the reference number in line 305 needs to be bolded.

And please double-check the manuscript for typos, e.g., the reference number in line 305 needs to be bolded.

Author Response

We thank the reviewer for the positive comments. Reference 124 in Line 305 is marked in boldface. The manuscript has been revised to correct English and typos. The corrections have been marked in blue.

Reviewer 4 Report

Dear authors,

Thank you for having addressed my comments (even though I don't know why you re-phrased them in your answers).

Minor checks before publication.

Author Response

We express our gratitude to the reviewer for their valuable contribution in enhancing the manuscript. The sentence in the abstract has been rephrased to align with your suggestion, emphasizing both the process (irregular and rapid contraction of the atria) and its effects (cardiac dysfunction and CVD sequelae). The manuscript has been thoroughly revised, and any English and typographical errors have been identified and marked in blue.